# Ambiguous Contribution of Glucocorticosteroids to Acute Neuroinflammation in the Hippocampus of Rat

**DOI:** 10.3390/ijms241311147

**Published:** 2023-07-06

**Authors:** Liya V. Tret’yakova, Alexey A. Kvichansky, Ekaterina S. Barkovskaya, Anna O. Manolova, Alexey P. Bolshakov, Natalia V. Gulyaeva

**Affiliations:** 1Institute of Higher Nervous Activity and Neurophysiology, Russian Academy of Sciences, 117485 Moscow, Russiaanna.manolova@ihna.ru (A.O.M.);; 2Research and Clinical Center for Neuropsychiatry of Moscow Healthcare Department, 115419 Moscow, Russia

**Keywords:** hippocampus, neuroinflammation, bacterial lipopolysaccharide, glucocorticosteroids, dexamethasone, mifepristone, spironolactone, microglia

## Abstract

Effects of modulation of glucocorticoid and mineralocorticoid receptors (GR and MR, respectively) on acute neuroinflammatory response were studied in the dorsal (DH) and ventral (VH) parts of the hippocampus of male Wistar rats. Local neuroinflammatory response was induced by administration of bacterial lipopolysaccharide (LPS) to the DH. The modulation of GR and MR was performed by dexamethasone (GR activation), mifepristone, and spironolactone (GR and MR inhibition, respectively). Experimental drugs were delivered to the dentate gyrus of the DH bilaterally by stereotaxic injections. Dexamethasone, mifepristone, and spironolactone were administered either alone (basal conditions) or in combination with LPS (neuroinflammatory conditions). Changes in expression levels of neuroinflammation-related genes and morphology of microglia 3 days after intrahippocampal administration of above substances were assessed. Dexamethasone alone induced a weak proinflammatory response in the hippocampal tissue, while neither mifepristone nor spironolactone showed significant effects. During LPS-induced neuroinflammation, GR activation suppressed expression of selected inflammatory genes, though it did not prevent appearance of activated forms of microglia. In contrast to GR activation, GR or MR inhibition had virtually no influence on LPS-induced inflammatory response. The results suggest glucocorticosteroids ambiguously modulate specific aspects of neuroinflammatory response in the hippocampus of rats at molecular and cellular levels.

## 1. Introduction

Neuroinflammation is a complex response of the central nervous system (CNS) to various stimuli, both internal and external ones, in normal or pathological situations. For example, neurogenesis and synaptic processes induce neuroinflammatory reaction as well as injuries, infections, and toxic substances [1]. Neuroinflammation contributes to cognitive impairments, neuronal cell loss, and microglial activation [2]. Microglia is believed to be a key player in neuroinflammation. These innate immune cells execute macrophage-like activities in the CNS and change their shape and expression profile during neuroinflammatory response [3]. Neuroinflammation is accompanied by release of cytokines and other paracrine factors (reactive oxygen and nitrogen species, chemokines, prostaglandins) as well as attraction of immune cells from peripheral bloodstream through the blood–brain barrier [1]. Administration of bacterial lipopolysaccharide (LPS) is routinely used to induce neuroinflammation in different studies [2,4,5,6]. Simplistically, LPS binds to Toll-like receptor 4 (TLR4) that leads to the signal transduction to specialized intracellular proteins, ultimately resulting in production of various inflammatory cytokines (for example, IL1β, IL6, TNFα) [2,4,5,6].

Glucocorticosteroids (GCS) are the main regulators of stress and inflammatory responses both at local and systemic levels. Glucocorticoid and mineralocorticoid receptors (GR and MR, respectively), two main types of GCS receptors, mediate their effector functions and are expressed by numerous cell types including cells of the nervous and immune systems. GCS are considered to be robust regulators of neuroinflammatory processes [7]. This may be exceptionally important for the hippocampus, a brain structure selectively vulnerable to neuroinflammation, at least partially, due to high density of GR and MR. High expression of these receptors predisposes hippocampal neurons to the damage that may be induced by out-of-control neuroinflammatory process. This damage may result in the development of hippocampus-associated pathologies, such as depressive spectrum disorders, epilepsy, and post-stroke and post-traumatic affective and cognitive impairments [8,9]. Modulation of GR and MR function may be a rational approach to study mechanisms of GCS involvement in neuroinflammatory response.

The association of GCS with inflammation is vigorously discussed. GCS are known for a long time as anti-inflammatory agents, which can effectively suppress peripheral immune responses [7]. However, there are studies reporting that chronic or acute systemic GCS administration as well as an increase in their production in the body may enhance inflammatory processes. For example, stronger proinflammatory response to either systemic [10] or local bacterial LPS administration [11] was demonstrated after stress, while pretreatment with GR antagonist mifepristone (MIF) or adrenalectomy abolished this effect. Peripheral corticosterone injection 2 or 24 h before systemic LPS administration resulted in proinflammatory responses to LPS in both peripheral macrophages and hippocampal microglia while administration of GCS after systemic LPS exposure suppressed the proinflammatory response to LPS [12]. Obviously, the effects of GCS may depend on the mode of their delivery. Intranasal administration of GR agonist dexamethasone (DEX) prevented systemic LPS-induced neuroinflammatory response, while intravenous injection of DEX at similar dose had an opposite effect [13]. Another group showed that DEX administration into the dentate gyrus enhanced neurogenesis, though systemic DEX injection suppressed this process [14].

It should be taken into account that powerful and diverse systemic influences of GCS significantly complicate the detection and understanding of their direct effects in the brain. To induce neuroinflammation locally and specifically and minimize vague reactions, a model of intrahippocampal administration of bacterial LPS was used. Using this approach, it is possible to study direct effects of GCS on the progress of neuroinflammation in brain tissue [15].

The aim of this study was to evaluate the effect of modulation of GR and MR on the acute local LPS-induced neuroinflammatory response in the hippocampus of rats at the molecular and cellular levels. We studied changes in the expression of neuroinflammation-related genes and the state of microglia after local injections of GR agonist (DEX), GR antagonist (MIF), or MR antagonist (spironolactone, SPIR) during LPS-induced neuroinflammation in the hippocampus.

## 2. Results

### 2.1. Experiment 1: Effects of GR Activation by DEX on LPS-Induced Neuroinflammation

In the first experiment, we evaluated the effect of GR activation on the markers of neuroinflammation and microglial morphology under control conditions and after LPS administration. However, before examining effects of GR activation, we performed a comparative study of the effects of intrahippocampal administration of PBS and LPS.

#### 2.1.1. Model Validation

##### Effects of PBS and LPS Administration on mRNA Expression of Neuroinflammation-Associated Genes

In Experiment 1, PBS injection did not produce significant effect on mRNA expression of the majority of the genes studied (*Il1b, Il6, Tnf, Tgfb1, Cx3cl1, Cx3cr1, Ncf1*) in the dorsal (DH) and ventral (VH) parts of the hippocampus as compared to the intact controls (Figure 1 and Appendix A). The absence of changes in the expression of these key cytokines points to the absence of acute inflammation 3 days after PBS administration. A significant increase in *Ccl2* expression level in the DH (Figure 1e) after PBS injection as compared to the Intact group in this experiment may reflect a residual inflammatory response.

LPS administration induced neuroinflammatory response in the DH detected as a significant increase in the expression levels of *Il1b* (Figure 1a), *Il6* (Figure 1b), *Tnf* (Figure 1c), *Tgfb1* (Figure 1d), *Ccl2* (Figure 1e), and *Ncf1* (Appendix A) as compared to the PBS-injected group. Regarding the fractalkine system, a significant increase in *Cx3cr1* expression (Appendix A) but no changes in *Cx3cl1* expression were observed in the DH (Appendix A).

##### Effects of PBS and LPS Administration on Microglial Activation

The state of microglia was evaluated based on its morphology. First, we counted the number of different morphological types of microglia. Second, we evaluated the percentage of Iba1-positive immunostained area (PIA). Activated microglia has enlarged soma and thickened primary processes, and microglia activation is reflected by an increase in PIA.

Evaluation of different subtypes of microglia showed that PBS administration induced only minor effect in the DH as compared to the Intact group. A trend of an increase in the number of ramified cells was observed (Figure 2 and Appendix A). After LPS injection, we found a trend of decrease in the number of ramified cells and a trend of increase in the number of amoeboid, rod, and symplast cells in the DH as compared to the PBS group (Figure 2 and Appendix A), and these alterations reflected microglial activation. In the VH, microglial cells mostly with resting morphology were noticed; a trend of increase in the number of rod microglia after LPS administration was also detected (Figure 2 and Appendix A).

Next, to validate the method of evaluation of microglia activation using PIA, we compared this parameter in the DH and the VH for each experimental group. In the DH, which is closer to injection site, there was a trend of increase in PIA in all experimental groups except for the Intact group (Figure 3). This fact indicated that each treatment, including PBS, induced changes in the state of microglial cells. We also detected an increase in PIA in PBS groups as compared with Intact groups in Experiments 1 and 2. Thus, the injection per se led to an activation of microglia in the DH (300 µm distant to the injection site) 72 h after injection.

Taken together, the above data on gene expression and microglial morphology suggest that the model of LPS-induced neuroinflammation can be used as a model of local neuroinflammation in the DH.

#### 2.1.2. Effects of DEX Administration on mRNA Expression of Neuroinflammation-Associated Genes under Control and Neuroinflammatory Conditions

DEX administration alone had only a minor effect on the expression of the genes studied. We detected trends of increase in the expression levels of *Il1b* (Figure 1a), *Il6* (Figure 1b), and *Ccl2* (Figure 1e) in the DH. However, when applied together with LPS, DEX attenuated the increase in the expression of selected cytokines induced by LPS. Significant decreases in the expression of *Il6* (Figure 1b) and *Ccl2* (Figure 1e) in the DH were observed after DEX+LPS administration as compared to the LPS group.

#### 2.1.3. Effects of DEX Administration on Microglial Activation under Control and Neuroinflammatory Conditions

Activation of GR also had weak proinflammatory effect on the state of microglia in the DH (increase in the number of hypertrophic cells and a trend of decrease in the number of ramified cells as compared to PBS-injected animals) (Figure 4a and Appendix A) but not in the VH, where microglia predominantly remained in the resting ramified state (Appendix A). DEX administration did not prevent changes in microglial morphology resulting from LPS-induced neuroinflammation in the DH either (Figure 4b and Appendix A). In the VH, mostly microglial cells with resting morphology were observed (Appendix A).

In this experiment, both groups administered with LPS (LPS and DEX/LPS) showed trends of increase in PIA as compared to the corresponding control groups (PBS and DEX, respectively) (Figure 3). There was no difference between the LPS and DEX/LPS groups either (Figure 3). According to these results, LPS induced an increase in PIA 3 days after treatment independent of DEX administration.

#### 2.1.4. Sucrose Preference Test after GR Activation

A significant decrease in sucrose preference was detected in the group of animals after combined administration of DEX and LPS on the first day after surgery as compared to respective animals before surgery (Figure 5).

### 2.2. Experiment 2: Effects of GR and MR Inhibition on LPS-Induced Neuroinflammation

In the second experiment, we evaluated the effects of GR and MR inhibition on selected parameters under basal and neuroinflammatory conditions.

#### 2.2.1. Model Validation

##### Effects of PBS and LPS Administration on mRNA Expression of Neuroinflammation-Associated Genes

Analysis of effects of PBS and LPS administration showed that their influence on the expression of inflammation-associated genes (Figure 6 and Appendix A) and microglial morphology (Appendix A) was similar to respective alterations demonstrated in the previous experiment. PBS injection induced a significant increase in *Ccl2* expression level in the DH (Figure 6e) and *Tgfb1* expression level in the DH (Figure 6d) and the VH (Appendix A) as compared to the Intact group. Similar to Experiment 1, LPS administration resulted in a significant increase in the expression levels of *Il1b* (Figure 6a), *Il6* (Figure 6b), *Tnf* (Figure 6c), *Tgfb1* (Figure 6d), *Ccl2* (Figure 6e), and *Ncf1* (Appendix A) in the DH, and *Il1b* (Appendix A) and *Ncf1* (Appendix A) in the VH as compared to the PBS-injected group.

The only apparent difference from Experiment 1 was the absence of LPS effect on the expression of fractalkine receptor gene *Cx3cr1* (Appendix A). The seeming lack of significant changes in *Cx3cr1* expression level (Appendix A) as compared to Experiment 1 was related to a higher dispersion of the data in the PBS group in Experiment 2. Obviously, in Experiment 1, the Intact and PBS-treated groups did not differ in the expression of this gene, and its dispersion was comparable, while in Experiment 2, the dispersion of expression of *Cx3cr1* gene in PBS group was much higher, resulting in apparent difference in the effect of LPS. However, as compared with Intact animals, LPS did exert an effect on the expression of this fractalkine receptor gene similar to that in Experiment 1 (Experiment 2, Intact vs. LPS: *Cx3cr1*, *p* = 0.00031).

##### Effects of PBS and LPS Administration on Microglial Activation

After PBS injection, microglial morphology did not show significant changes either in the DH (Figure 2b and Appendix A) or the VH (Figure 2d and Appendix A). Similar to Experiment 1, we observed more microglial cells with disease-associated (DAM) phenotype in the DH (Figure 2c and Appendix A) and with resting phenotype in the VH (Figure 2d and Appendix A) after LPS administration. Further analysis using PIA showed that PBS induced significant activation of microglia, similar to that in Experiment 1 (Appendix A).

#### 2.2.2. Effects of MIF and SPIR Administration on mRNA Expression of Neuroinflammation-Associated Genes under Control and Neuroinflammatory Conditions

Administration of GR inhibitor MIF or MR inhibitor SPIR induced a trend of decrease in the expression of *Il1b* (Figure 6a) and *Tgfb1* (Figure 6d) in the DH and did not affect the expression of other genes studied in either the DH or the VH (Figure 6 and Appendix A). Under proinflammatory conditions, inhibition of GR and MR had a very weak effect. We only revealed a trend of a decrease in expression of *Il6* in the DH (Figure 6b and Appendix A) after combined MIF+LPS injection.

#### 2.2.3. Effects of MIF and SPIR Administration on Microglial Activation under Control and Neuroinflammatory Conditions

Consistent with gene expression levels, MIF and SPIR did not induce alterations in microglial phenotype as compared to PBS-injected animals (Appendix A). We found that neither SPIR nor MIF alone could change PIA (Appendix A). Inhibition of either GR or MR during LPS application did not affect microglial morphology, promoting the DAM phenotype of microglial cells (Appendix A). We also did not find any significant differences between LPS-injected groups (LPS, MIF+LPS, and SPIR+LPS) and corresponding control groups (PBS, MIF and SPIR, respectively) (Appendix A).

#### 2.2.4. Sucrose Preference Test after GR and MR Inhibition

Analysis of behavioral effects in the sucrose preference test could not reveal significant effects of GR and MR inhibitors either alone or in combination with LPS on the hedonistic behavior (Appendix A).

## 3. Discussion

The aim of this study was to analyze the effects of GR and MR modulation on the neuroinflammatory response in the hippocampus of rats. We evaluated direct effects of GR agonist DEX, GR antagonist MIF, and MR antagonist SPIR on the hippocampal tissue by changes in expression levels of neuroinflammation-related genes and in morphology of microglia. We used intrahippocampal injection to avoid systemic influence of experimental drugs. Their direct effects on brain tissue were evaluated on the third day after surgery. The intrahippocampal administration of PBS had a very weak local inflammatory effect, which disappeared by this time.

In the periphery, GCS are well-known anti-inflammatory substances, which realize their functions by binding to GR and MR; however, in the brain, GCS can be either anti-inflammatory or proinflammatory agents [16]. Their effects depend on a number of factors, such as tissue conditions [12] and the mode of their administration [13,14]. It is believed that in normal conditions endogenous GCS are present in the brain, bind to GR and MR and thus realize their pleiotropic functions. In this study, activation of GR by DEX resulted in a weak proinflammatory shift in the expression of cytokines in both the dorsal and ventral parts of the hippocampus. The predominance of activated forms of microglia in the DH also confirmed the development of neuroinflammation. Proinflammatory effects of GCS at high concentrations were previously demonstrated in several studies [17,18,19], and we recently extended previous findings by showing that direct application of DEX to the hippocampal tissue had similar proinflammatory effect [20].

Our data on the expression of proinflammatory cytokines showed that DEX induced an increase in mRNA level of some of them (*Il6, Ccl2, Tnf*), whereas combined application of DEX and LPS suppressed inflammatory response. At first glance, these findings look quite paradoxical; however, many studies showed that the modality of effects of GCS on neuroinflammation may be variable depending on the experimental conditions [16]. It was shown that proinflammatory effect of GCS in either brain tissue or cultured macrophages was evident when a treatment with GCS preceded proinflammatory stimuli, whereas, when applied after proinflammatory stimuli, GCS demonstrated anti-inflammatory effects [12]. In experiments of our present study, two factors were interacting, tissue damage by implanted needle and DEX. Tissue damage inevitably leads to inflammation due to release of damage-associated molecular patterns (DAMPs) by injured cells and surrounding macrophages. Presumably, the time course of DEX effect and DAMP-induced neuroinflammation is different, the latter being much slower. Therefore, under these conditions, DEX may serve as an agent priming microglia and enhancing slowly developing proinflammatory response, similar to the results described in [12]. However, in the presence of DEX, LPS may induce a more rapid proinflammatory response since microglia express respective receptors, and the effect of DEX appears to be anti-inflammatory, as described in previous studies [16]. Moreover, it was shown that LPS induces an exposure of membrane-associated GR on the membrane of macrophages [21], which may be one of factors shifting the effect of GCS to anti-inflammatory.

In contrast to DEX, inhibition of GR and MR by MIF and SPIR, respectively, did not significantly affect the expression of cytokines and microglial morphology. This result suggests that under basal conditions or conditions of weak inflammation evoked by brain microtrauma in the area of injection, GCS do not induce expression of proinflammatory cytokines and respective microglial morphology. Thus, under basal conditions, GCS do not appear to be critical hormones mediating neuroinflammatory signaling.

Further, we evaluated effects of GR and MR modulation on LPS-induced neuroinflammatory response in the hippocampus by co-injection of DEX, MIF, or SPIR together with LPS. Previous studies reported that LPS alone induced neuroinflammation and the appearance of activated microglial cells [22,23,24,25,26]. Indeed, our study demonstrated an increase in expression of neuroinflammation-associated genes (*Il1b*, *Il6*, *Tnf*, *Tgfb1*, *Ccl2*, and *Ncf1*) in the DH. After induction of neuroinflammation in the hippocampus by LPS, we expected that effects of activation and inhibition of GR and MR would be more expressed than in the basal conditions. DEX suppressed expression of *Il6*, *Ccl2,* and fractalkine (*Cx3cl1*), confirming anti-inflammatory properties of DEX. At the cellular level, DEX generally did not affect the expression of damage-associated microglia appearing after LPS treatment; however, DEX induced the appearance of specific, qualitatively different morphological microglial changes as compared to separate administration of DEX or LPS. In contrast, inhibition of GR and MR activity during LPS-induced neuroinflammation had a very weak effect on either expression of cytokines or microglial morphology. This may indicate that the development of cytokine-mediated cellular response to LPS-induced neuroinflammation is not significantly modulated by endogenous GCS presumably released during the generalized inflammatory process in the organism. However, extra activation of GR by exogenous agent DEX can modulate the neuroinflammation, suggesting that GR are involved in the regulation of some routes of neuroinflammatory process related to fractalkine- and IL6-mediated signaling. These facts confirm the complex nature of neuroinflammation and suggest that GCS may control all facets of this process. Thus, we observe that GCS contribution to the neuroinflammatory process in the hippocampus is really ambiguous and complex. The lack of unambiguous data on this issue prevents from making clear recommendations for the use of GCS in neuroinflammation.

## 4. Materials and Methods

### 4.1. Animals

The study was performed on 89 male Wistar rats (300–400 g). The animals were purchased in the “Stolbovaya” Breeding Center (Moscow, Russia) and were housed individually in the cages in the isolated vivarium. The rats were kept at 12 h: 12 h light/dark cycle, with free access to food and water.

### 4.2. Experimental Protocol and Surgery

In this study, DEX (0.05 g/L; D1756, Sigma, USA), SPIR (5.4 g/L; S3378, Sigma, New York, NY, USA), and MIF (4.5 g/L; M8046, Sigma, USA) were used to modulate the functioning of GR and MR. Bacterial LPS *E. coli* (0.2 g/L; serotype O26:B6, Sigma-Aldrich, St. Louis, MO, USA) was injected into the DH for induction of acute local neuroinflammation. This LPS dose was selected because pilot studies indicated that this dose of LPS induces moderate proinflammatory response in the hippocampus. The phosphate-buffered saline (PBS) was used as a vehicle for all experimental substances. The concentrations of GR and MR modulators were chosen based on EC_50_ and IC_50_ values of the drugs for GR and MR [27] and corresponded to approximately tenfold EC50 or IC50 when adjusted to the volume of the DH (~50 µL).

The study included two separate experiments.

In Experiment 1, effects of GR agonist DEX on the development of acute neuroinflammation in the hippocampal tissue were studied. The animals were divided into five groups, eight rats per group: Intact group without manipulations, PBS control group, LPS group, DEX group, and DEX+LPS group.

In Experiment 2, the effect of GR and MR inhibitors (MIF and SPIR, respectively) on the development of acute neuroinflammation in the hippocampal tissue was explored. The animals were divided into seven groups, six to eight rats per group: Intact group without manipulations, PBS control group, LPS group, MIF group, MIF+LPS group, SPIR group, and SPIR+LPS group. The rats of Intact groups of both experiments did not undergo surgery; they were only tested for sucrose preference until decapitation as explained in the respective section of Materials and Methods below.

Thus, DEX, MIF, and SPIR were injected either alone (basal conditions) or in combination with LPS (neuroinflammatory conditions). The mixtures of drugs with LPS were injected through the same syringe. For each substance administered, the total volume of injected solution was 1 µL per hippocampus.

All surgical procedures were performed under inhalation anesthesia with 2–3% isoflurane. Rats were fixed in a stereotactic frame, then scull trepanations were created in the left (AP= −3.5 mm, L= +2.00 mm) and right (AP= −3.5 mm, L= −2.00 mm) parietal bones. Experimental substances (1 µL) were delivered to the dentate gyrus of the DH bilaterally (H= −3.5 mm) with a Hamilton syringe at the injection rate of 0.15 µL/s.

After the surgery, the animals were returned to their home cages. Three days later, the rats were anesthetized with 10% chloral hydrate and perfused transcardially with ice-cold 0.9% saline. The brains were removed and briefly cooled in ice-cold saline. The hippocampus was isolated, and its left part was divided into the DH and the VH for further analysis of expression of neuroinflammation-associated genes (*Il1b, Tnf, Il6, Tgfb1, Ccl2*) and microglial marker genes (*Cx3cl1, Cx3cr1, Ncf1*) using qPCR. The right hemisphere was fixed in 4% solution of formaldehyde for subsequent Iba1 immunohistochemical staining to examine the development of microgliosis.

To evaluate the degree of depressive-like behavior after GR and MR modulation in basal conditions and after induction of local neuroinflammation, we used the sucrose preference test. Reduced consumption of sucrose (anhedonia) in this test is considered as a reliable indicator of depressive-like behavior in rats [28].

The number of animals per group was calculated from the pilot series of experiments with the power goal set as 0.8. For the behavioral experiments, the minimum number of animals per group was 6.

The general design of Experiment 1 and Experiment 2 was similar and is shown in Figure 7.

### 4.3. RNA Extraction and Reverse Transcription

RNA was isolated using ExtractRNA reagent according to the manufacturer’s recommendations. Before cDNA synthesis, 2 μg of RNA was treated with DNase I (Thermo Fisher Scientific, Waltham, MA, USA) according to the manufacturer’s recommendations. Then, RNA was used to synthesize cDNA using a mix of random decaprimer (Evrogen, SB002, Moscow, Russia) and oligo(dT)-primer (Evrogen, SB001, Moscow, Russia) by means of the MMLV RT Kit (Evrogen, Moscow, Russia) in accordance with the manufacturer’s recommendations.

### 4.4. Quantitative Real-Time Polymerase Chain Reaction (qPCR)

The gene expression was analyzed using qPCRmix-HS SYBR+LowROX (Evrogen, PK156L, Russia) in accordance with the manufacturer’s recommendations by means of a quantitative PCR system CFX384 (Bio-Rad, Hercules, CA, USA). The genes of interest were *Il1b, Tnf, Il6, Tgfb1, Ccl2, Cx3cl1, Cx3cr1,* and *Ncf1*. Nucleotide sequences of primers used are shown in Appendix A.

The relative quantity (RQ) of transcripts was assessed using the 2^−ΔΔCt^ method, taking into account the efficiency of the reaction with respect to the expression of the *Hprt2* and *Ywhaz* genes; the data in the graphs are presented as relative quantity.

### 4.5. Histology and Morphometry

Sixty µm frontal sections of brain right hemispheres were prepared using a vibratome. The sections were placed in a cryoprotector solution (PBS:ethyleneglycol:glycerin, 1:0.75:0.75) and stored at −20 °C. Immunohistochemical staining for microglial marker Iba1 was performed using primary antibodies (rabbit anti-Iba1 IgG; Wako, Japan) diluted 1:400 with secondary antibodies (goat anti-rabbit IgG (H+L) Alexa-488, Invitrogen) diluted 1:500. The fixation of sections was performed using anti-fade histology mounting medium with DAPI (Sigma-Aldrich, St. Louis, MO, USA). Microphotographs of immunohistochemically stained sections were prepared using a Keyence BZ-9000 microscope (×20). Further calculations of microglial cells and image processing were performed using ImageJ/Fiji software (Version 1.53c, NIH, LOCI, University of Wisconsin, USA). The three rectangular areas were cropped from each image gained from microscope. It was performed to ensure the absence of artifacts and the location of analyzed area in certain single layer of hippocampus (namely, radial layer of CA1 hippocampal subfield). The latter is important since the features and morphology of microglial cells differ between different brain areas and even between hippocampal layers. The number of microglial cells was counted in each section of the three fields 100 × 100 µm in the stratum radiatum in CA1 area and summarized. We used three sections from each animal; the sections included rostral DH (located +250–300 µm from the injection site), caudal DH (−250–300 µm from the injection site), and the VH with the CA1 field. The results from rostral and caudal parts of the DH were averaged because of similar morphological picture. The selection of microscopy fields and microglia counts were performed blinded to the experimental conditions.

During the counting, we distinguished the following five subtypes of microglial cells [29]:(1)Ramified microglia, resting phenotype of microglia, which are characterized by small roundish soma, highly branched and thin processes, and roundish nucleus that occupies almost the entire space of the soma;(2)Amoeboid microglia, morphological phenotype, which are very similar to macrophages with large round cell bodies without processes;(3)Hypertrophic microglia, functionally activated phenotype of microglia characterized by enlarged soma with shorter and thicker processes as compared to ramified microglia. The nuclei of hypertrophic microglia do not occupy the entire soma;(4)Rod microglia, microglial phenotype, which are characterized by elongated cells with elongated nuclei;(5)Symplasts, cells with morphology similar to rod subtype but with several nuclei.

The subtypes (2)–(5) are related to disease-associated microglia (DAM phenotype). Representative photos for different microglial subtype are shown in Figure 8.

Additionally, analysis of microglia activation state was performed as quantification of the percentage of Iba1-positive immunostained area (PIA).Cropped images were split by channels, and only channel with Iba-1 staining was processed further. The following steps were performed to evaluate the optical density of the immunohistochemically stained tissue: (1) The background correction; (2) The contrast enhancement; (3) Binarization (with the Yen filter). Then, the percent of white pixels (e.g., belonging to positively immunostained tissue) on the cropped image were calculated. Here is the ImageJ script that was used for the processing of each cropped image:run(“Split Channels”);selectWindow (filename + “Iba”);run(“Subtract Background…”, “rolling = 50”);run(“Enhance Contrast…”, “saturated = 0.3 normalize”);run(“Auto Threshold”, “method = Yen white”);run(“Set Scale…”, “distance = 0 known = 0 unit = pixel global”);run(“Set Measurements…”, “area_fraction redirect = None decimal = 3”);run(“Measure”);

For subsequent statistical evaluation, all the data were averaged per animal.

### 4.6. Sucrose Preference Test

Rats were adapted on day 1 to their individual cages where they were placed for conducting the sucrose preference test. During the following 5 days, the rats were given a choice between drinking bowls with fresh water and 1% sucrose solution in water ad libitum. Both bowls were weighted every day during the sucrose preference test, and the amount of water and sucrose solution was recorded (g). The index of sucrose preference (SP, %) was calculated according to the formula:(1)SP i=m (sucrose, i)msucrose, i+m (water,i)Me (msucrose, intactmsucrose, intact+mwater, intact)∗100%,
where *m* is weight of the bowl (with water or sucrose solution); *i*, any animal of following experimental groups (PBS, LPS, DEX, DEX+LPS, MIF, MIF+LPS, SPIR, SPIR+LPS); *Me*, median; intact, any animal of Intact group.

### 4.7. Statistical Analysis

The variable values were not normally distributed, and groups had different variances (according to Shapiro–Wilk and Levene tests). Outliers were removed automatically according to 1.5 IQR method. The significance of differences between the experimental groups was determined using the Kruskal–Wallis (KW) H test with Mann–Whitney post hoc test with multiple comparison (Bonferroni) correction in the Python 3.9 scipy 1.7.1 software package: (1) five hypotheses for results of the 1st phase, α = 0.01 (we performed following comparisons of groups: Intact vs. PBS, PBS vs. LPS, PBS vs. DEX, DEX vs. DEX+LPS, LPS vs. DEX+LPS); (2) Eight hypotheses for results of the 2nd phase, α = 0.00625 (we performed following comparisons of groups: Intact vs. PBS, PBS vs. LPS, PBS vs. SPIR, PBS vs. MIF, SPIR vs. SPIR+LPS, MIF vs. MIF+LPS, LPS vs. SPIR+LPS, LPS vs. MIF+LPS). The significances of experimental effects on sucrose preference were determined by the Friedman test with Nemenyi post hoc test in the Python 3.9 scipy 1.7.1 and scikit_posthocs 0.7.0 software packages.

The data on graphs are presented as box plots: bars represent medians, boxplots represent quartiles, and whiskers represent Q1 − 1.5 × IQR and Q3 + 1.5 × IQR.

## 5. Conclusions

In conclusion, local GR activation by DEX administration had a very weak proinflammatory effect on the hippocampal tissue in normal conditions (trends of increase in *Il1b*, *Il6*, and *Ccl2* gene expression). During LPS-induced neuroinflammation, GR activation by DEX suppressed expression of proinflammatory cytokines (decrease in *Il6* and *Ccl2* gene expression) but maintained activated state of microglia. Unexpectedly, inhibition of GR and MR by intrahippocampal injections of either MIF or SPIR did not exert expected opposite effects against GR activation by DEX. Our results suggest that the GCS ambiguously modulate specific aspects of neuroinflammatory response in the hippocampus on both molecular and cellular levels. Directly or indirectly, both GCS and neuroinflammatory processes are involved in virtually all mechanisms of hippocampal neuroplasticity [30], and the routes of this control by GCS are obviously much more intricate than what we can hypothesize based on available but yet obviously incomplete experimental data.

### Limitations of the Study

One of obvious limitations of our study is the lack of measurements of proteins levels of proinflammatory cytokines. It is well known that damage of CNS tissue or LPS injection led to elevation of inflammatory cytokine expression, and in this study, we used the changes in the levels of mRNA of various proinflammatory factors as manifestations of inflammatory process intensity. This approach allowed us to detect effects on neuroinflammation of intrahippocampal injections of various drugs. Undoubtedly, measurements of protein levels of cytokines may provide additional important data on the modulation of neuroinflammatory processes. However, it is worth to note that cytokine lifetime in the tissue is quite a sophisticated issue, and frequently, changes in mRNA of cytokines are not necessarily associated with respective changes in protein levels (see, for example, [31,32]). Moreover, the absence of changes in the protein level does not necessarily mean that the changes in the mRNA level, which we have observed in the current study, are not functional. Moreover, an absence of changes may reflect an intensification of cytokine proteins turnover in inflammatory conditions. To prove the latter suggestion, an additional and very thorough investigation should be performed.

## Figures and Tables

**Figure 1 ijms-24-11147-f001:**
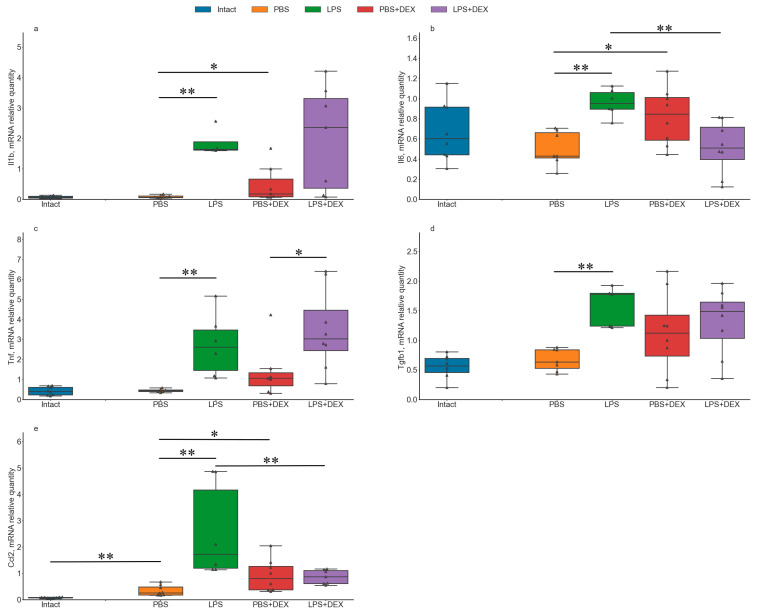
mRNA expression levels of neuroinflammation-associated genes *Il1b* (**a**), *Il6* (**b**), *Tnf* (**c**), *Tgfb1* (**d**), and *Ccl2* (**e**) in the DH after intrahippocampal administration of PBS, LPS, DEX, or DEX+LPS. N =8. *Il1b,* KW: H = 13.5849, *p* = 0.0035; *Il6,* KW: H = 13.2089, *p* = 0.0042; *Tnf,* KW: H = 14.3247, *p* = 0.0025; *Tgfb1,* KW: H = 8.1057, *p* = 0.0439; *Ccl2,* KW: H = 16.5747, *p* = 0.0009. * and **, a trend (0.01 ≤ *p* ≤ 0.05) and significant (*p* < 0.01) differences, respectively, according to Mann–Whitney U-test with multiple comparison correction.

**Figure 2 ijms-24-11147-f002:**
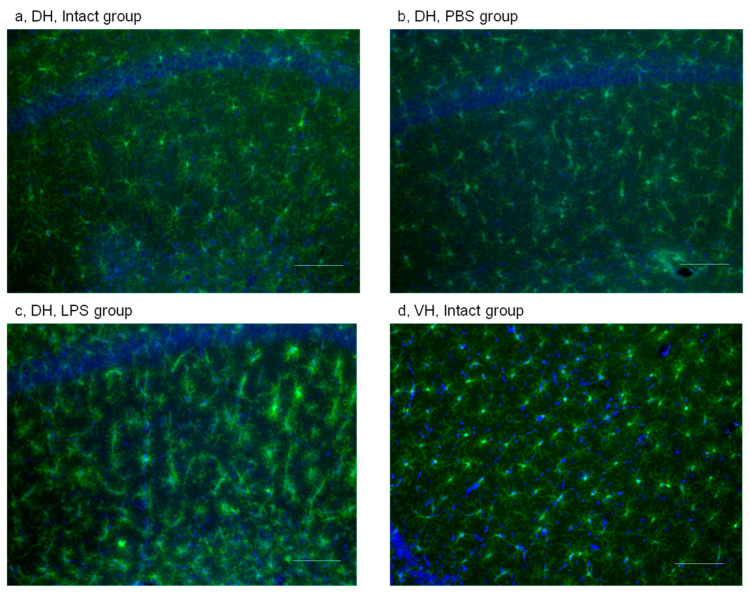
Microglial activation in the hippocampus with representative microphotographs. (**a**) DH Intact group; (**b**) DH PBS group; (**c**) DH LPS group; (**d**) VH Intact group. Microglial staining in the VH of PBS and LPS groups are similar to the VH of Intact group. Staining with anti-Iba1 and DAPI, ×20. Scale bar 100 µm.

**Figure 3 ijms-24-11147-f003:**
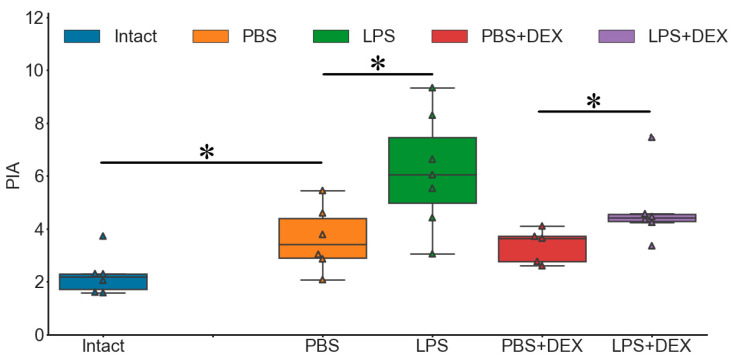
The percentage of Iba1-positive immunostained area (PIA) in the DH after intrahippocampal injection of PBS, LPS, DEX, or DEX+LPS, n = 7. PIA, KW: H = 9.6055, *p* = 0.0222. *, differences at the trend level (0.01 ≤ *p* ≤ 0.05) according to Mann–Whitney U-test with multiple comparison correction.

**Figure 4 ijms-24-11147-f004:**
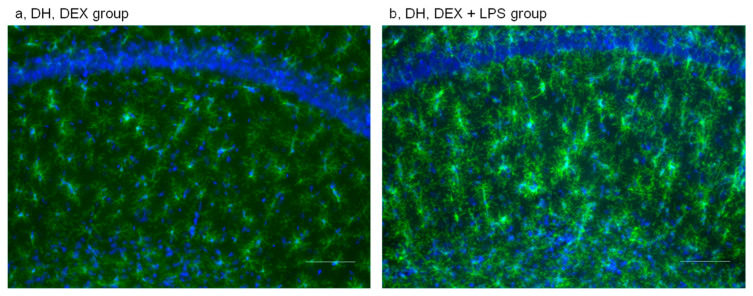
Microglial activation in the DH with representative microphotographs. (**a**) DH DEX group; (**b**) DH DEX+LPS group. Staining with anti-Iba1 and DAPI, ×20. Scale bar 100 µm.

**Figure 5 ijms-24-11147-f005:**
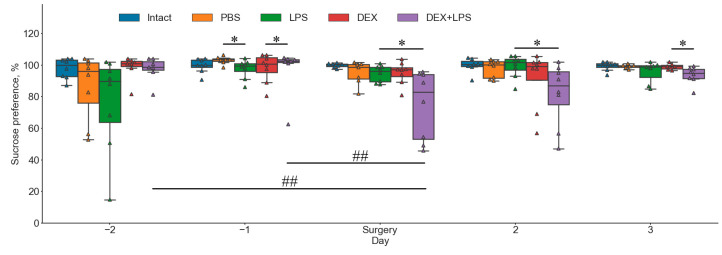
Changes in sucrose preference (%) of individual animals before and after intrahippocampal administration of PBS, LPS, DEX, or DEX+LPS, n = 8. *, differences at the trend level (0.01 ≤ *p* ≤ 0.05) between groups on the same day, according to Mann–Whitney U-test with multiple comparison correction. ##, significant (*p* < 0.01) differences between animals within the same group at different time points, according to Friedman test with Nemenyi post hoc test.

**Figure 6 ijms-24-11147-f006:**
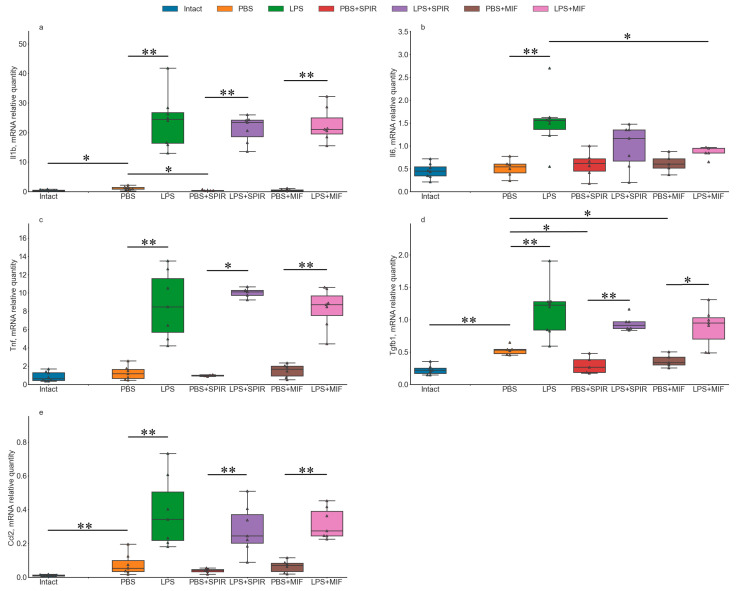
mRNA expression levels of neuroinflammation-associated genes *Il1b* (**a**), *Il6* (**b**), *Tnf* (**c**), *Tgfb1* (**d**), and *Ccl2* (**e**) in the DH after intrahippocampal injection of PBS, LPS, SPIR, SPIR+LPS, MIF, or MIF+LPS. Intact and LPS groups, n = 8; PBS, SPIR+LPS, and MIF+LPS groups, n = 7; SPIR and MIF groups, n = 6. *Il1b,* KW: H = 29.1944, *p* = 0.00002; *Il6,* KW: H = 15.5014, *p* = 0.0084; *Tnf,* KW: H = 26.8163, *p* = 0.00006; *Tgfb1,* KW: H = 26.1162, *p* = 0.00008; *Ccl2,* KW: H = 27.6204, *p* = 0.00004. * and **, a trend (0.00625 ≤ *p* ≤ 0.05) and significant (*p* < 0.00625) differences, respectively, according to Mann–Whitney U-test with multiple comparison correction.

**Figure 7 ijms-24-11147-f007:**
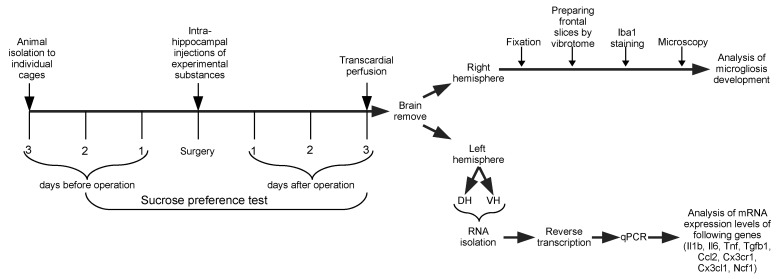
Experimental design.

**Figure 8 ijms-24-11147-f008:**
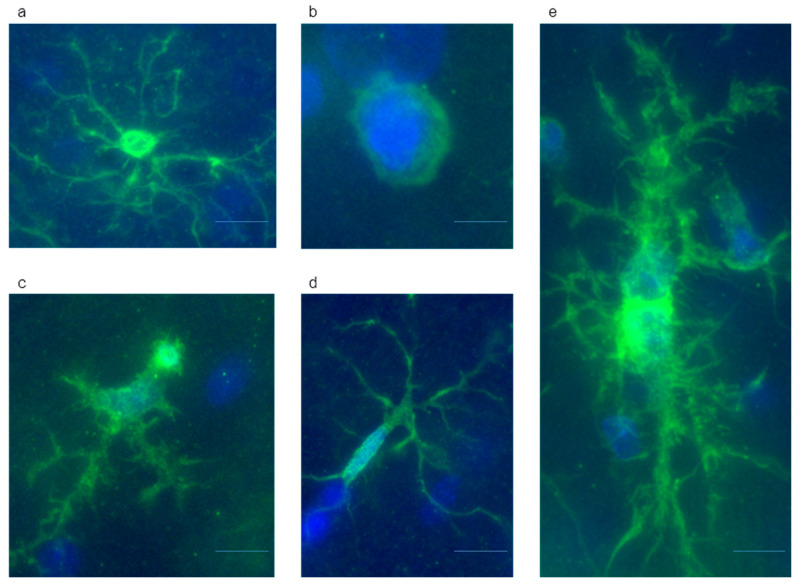
Representative microphotographs of microglial cells of different subtypes. (**a**) Ramified microglia; (**b**) Amoeboid microglia; (**c**) Hypertrophic microglia; (**d**) Rod microglia; (**e**) Symplasts. Staining with anti-Iba-1 andDAPI; immerse objective ×60. Scale bar 10 µm.

## Data Availability

Raw data are available from the authors upon reasonable request.

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
