# Peer review of "Ambiguous Contribution of Glucocorticosteroids to Acute Neuroinflammation in the Hippocampus of Rat"

_ijms, 2023, doi:10.3390/ijms241311147_

Round 1

Reviewer 1 Report (Previous Reviewer 3)

The revised version reads well. The use of the Statistics is made clear.

The supplemental data/figures are able to be viewed well. I hope the journal makes them full page figures.

This is an interesting study and I feel readers will enjoy reading the manuscript.

Author Response

The authors greatly appreciate the positive evaluation of this paper.

Reviewer 2 Report (Previous Reviewer 2)

No further comments.

Minor editing.

Author Response

The authors greatly appreciate the positive evaluation of this paper. Minor editing of English language required by the Reviewer has been performed

This manuscript is a resubmission of an earlier submission. The following is a list of the peer review reports and author responses from that submission.

Round 1

Reviewer 1 Report

The authors investigated about the effects of modulation of glucocorticoid and mineralocorticoid receptors on the acute local LPS-induced neuroinflammatory response in the hippocampus of rats.

Overall the topic could be interesting but some details could be improved.

 I recommend that the paper be accepted with minor revision:

a) The authors should mentioned in the abstract more details about model used.

b)   In the introduction section, little previous evidence is provided about the importance of neuroinflammation in daily life. Incorporating comparisons with other studies would increase the strength of the paper. Please refer to doi: 

10.3390/ani10050898; 10.1038/s41598-019-42286-8 10.3390/antiox10050818.  Additionally, more attention to how LPS induces inflammation and oxidative stress in other diseases should be added to improve this section. Please refer to doi:  10.3390/ijms22115533; 10.3892/etm.2019.8396; 10.3389/fncel.2020.00142; 10.3390/antiox9080693; 10.1177/0023677215570087. c)  The authors should clarify how they choose the number of animals. 

d)   The authors should better emphasize the conclusions.

e) There are some minor grammar issues that should be fixed in order to aid the accessibility of the results to the reader.

Author Response

The authors greatly appreciate the criticism and advice of the Referee and have revised the manuscript accordingly. All changes introduced are highlighted. 

Comments and Suggestions for Authors

The authors investigated about the effects of modulation of glucocorticoid and mineralocorticoid receptors on the acute local LPS-induced neuroinflammatory response in the hippocampus of rats.

Overall the topic could be interesting but some details could be improved.

  I recommend that the paper be accepted with minor revision:

  1. a) The authors should mentioned in the abstract more details about model used.

Response: We used stereotaxic injections to deliver experimental drugs to the dentate gyrus of the DH bilaterally. Thus, dexamethasone, mifepristone, and spironolactone were injected either alone (basal conditions) or in combination with LPS (neuroinflammatory conditions). Respective details have been added to the Abstract.

  1. b) In the introduction section, little previous evidence is provided about the importance of neuroinflammation in daily life. Incorporating comparisons with other studies would increase the strength of the paper. Please refer to doi: 10.3390/ani10050898; 10.1038/s41598-019-42286-8 10.3390/antiox10050818.

Additionally, more attention to how LPS induces inflammation and oxidative stress in other diseases should be added to improve this section. Please refer to doi:  10.3390/ijms22115533; 10.3892/etm.2019.8396; 10.3389/fncel.2020.00142; 10.3390/antiox9080693; 10.1177/0023677215570087.

Response: Respective changes have been made and additional references included.

  1. c) The authors should clarify how they choose the number of animals.

Response: The number of animals per group was calculated from the pilot series of experiments with the power goal set as 0.8. For the behavioural experiments, the minimum number of animals per group was 6. This information has been included to the text.

  1. d) The authors should better emphasize the conclusions.

Response: Respective changes have been made in the text.

  1. e) There are some minor grammar issues that should be fixed in order to aid the accessibility of the results to the reader.

Response: The text has been proofread.

Reviewer 2 Report

The manuscript “Ambiguous contribution of glucocorticosteroids to acute neuroinflammation in the hippocampus of rats” covers an interesting topic; however, the results need to be supported by additional studies.

Comments:

  1. Results could be presented better using separate sections for the different assays/techniques.
  2. The study includes the gene expressions of the various pro-inflammatory factors; however, it needs to be supported by protein expression studies.
  3. The levels of TNF alpha, IL6 etc., should be estimated in the brain tissue to support the results.
  4. The quantification of the fluorescence intensity should be done in the IHC.
  5. The IHC images should be included in the main manuscript and one or two figures, not supplementary.
  6. Why were there increases in cytokines expression in the DEXA group? Discuss properly.
  7. The variation in the expression of cytokines should be discussed with proper justification/reasoning.
  8. The response to the sucrose preference test should be appropriately discussed.
  9. Provide clear recommendations in conclusion on the use of GC in neuroinflammation.
  10. Many typos are there in the manuscript, and English editing is required.

Author Response

The authors greatly appreciate the criticism and advice of the Referee and have revised the manuscript accordingly. All changes introduced are highlighted. 

Comments and Suggestions for Authors

The manuscript “Ambiguous contribution of glucocorticosteroids to acute neuroinflammation in the hippocampus of rats” covers an interesting topic; however, the results need to be supported by additional studies.

Comments:

  1. Results could be presented better using separate sections for the different assays/techniques.

Response: Done

  1. The study includes the gene expressions of the various pro-inflammatory factors; however, it needs to be supported by protein expression studies.
  2. The levels of TNF alpha, IL6 etc., should be estimated in the brain tissue to support the results.

Response: We agree with the reviewer that protein expression may provide additional information about processess occuring in brain tissue under experimental conditions in our studies. However, we consider the changes in the levels of mRNA of various proinflammatory factors as recognized biological marker of inflammatory process. It is well known that damage of CNS tissue or LPS injection lead to elevation in expression of inflammatory cytokines and, in our opinion, it is not necessary to prove it again. Moreover, cytokine life time in brain tissue is quite a sophisticated issue and frequently in our hands changes in mRNA of cytokines are not necessarily associated with changes in protein level AT SAME TIME POINTS(see, for example, our studies https://doi.org/10.1007/s12035-021-02668-4; https://doi.org/10.1134/S0006297921060079). Since we used one time point in this study, the data on protein expression might not necessarily be in line with mRNA expression, thus introducing an unnecessary apparent contradiction of the data. Indeed, the absence of changes in the protein level does not mean that changes in the mRNA level were not functional: in addition to differences in time course in gene and protein expression, it may also reflect intensification of cytokine turnover during inflammation. To prove this an additional and very thorough investigation is needed which is outside of scope of our study. We believe that the use of the information about mRNA levels of cytokines is sufficient for our experiments since our aim was to analyze inflammatory markers to evaluate the intensity of inflammatory process but, at this stage, not to provide extended cytokine profile of inflammatory process induced by LPS or other substances used.

  1. The quantification of the fluorescence intensity should be done in the IHC.

Response: We performed IHC against Iba1 protein to analyze changes in the morphology of microglia after various treatments. Anti-Iba1 staining may be used to detect these changes. However, measurements of fluorescence intensity seem to be irrelevant for the analysis that we have performed since the quantitative relationship between the Iba1 protein level (or staining intensity) and the state of microglial activation is not direct but quite complex. We are afraid that it would be extremely speculative to make any conclusions on the basis of changes in the staining intensity and have avoided it purposely.

  1. The IHC images should be included in the main manuscript and one or two figures, not supplementary.

Response: Two IHC images from the Supplement have been moved to the main text.

  1. Why were there increases in cytokines expression in the DEXA group? Discuss properly.

Response: A large portion of discussion on this issue was included into the text.

  1. The variation in the expression of cytokines should be discussed with proper justification/reasoning.

Response: Unfortunately we did not fully understand this issue. If it is related to the effect of substances used on cytokine response, additional discussion has been introduced.

  1. The response to the sucrose preference test should be appropriately discussed.

We found the decrease in sucrose preference only in the group of animals after combined injection of DEX and LPS on the first days after the surgery i.e. this impact was so significant that we could see its effects on behavior (anhedonia). As compared single injections of LPS or DEX did not result in such effects. Probably, anhedonia is a reflection of strong neuroinflammatory process in the first days after DEX+LPS administration. We included the discussion into the text/

  1. Provide clear recommendations in conclusion on the use of GC in neuroinflammation.

Response: We see that GCS contribution to the neuroinflammatory process in the hippocampus is really ambiguous and complex. So there is not enough data yet on this issue to make clear recommendations for the use of GCS in neuroinflammation. We mentioned this in the revised text/

  1. Many typos are there in the manuscript, and English editing is required. Response: we did our best.

Reviewer 3 Report

I found this to be a very interesting study . The results of GR activation by DEX not altering  that majority of cytokines induced by LPS is fascinating. I would think this study would help better direct critical care with patients in septicemic states from gram-negative bacterial infections.

Minor edits:

1.       Line 154 “The only apparent difference was dissimilar LPS effect on the expression of genes 154 related to fractalkine system, Cx3cl1 (Figure S9b) and Cx3cr1 (Figure S9a).”

to

“The only apparent difference that was dissimilar LPS effect on the expression of genes 154 related to fractalkine system, Cx3cl1 (Figure S9b) and Cx3cr1 (Figure S9a).”

2.       Justify why 200 mg/l was used. Is this not a high concentration  compared to other reported studies ?

“Bacterial LPS E. coli (0.2 g/l; serotype O26:B6, Sigma-Aldrich, USA) was injected into 256 the dorsal hippocampus (DH) for induction of acute local neuroinflammation.”

Author Response

The authors greatly appreciate the criticism and advice of the Referee and have revised the manuscript accordingly. All changes introduced are highlighted. 

Comments and Suggestions for Authors

1). I found this to be a very interesting study. The results of GR activation by DEX not altering  that majority of cytokines induced by LPS is fascinating. I would think this study would help better direct critical care with patients in septicemic states from gram-negative bacterial infections.

Minor edits:

Line 154 “The only apparent difference was dissimilar LPS effect on the expression of genes 154 related to fractalkine system, Cx3cl1 (Figure S9b) and Cx3cr1 (Figure S9a).”

to

“The only apparent difference that was dissimilar LPS effect on the expression of genes 154 related to fractalkine system, Cx3cl1 (Figure S9b) and Cx3cr1 (Figure S9a).”

Response: Done

2). Justify why 200 mg/l was used. Is this not a high concentration  compared to other reported studies ?

Response: The following text is added to the revises manuscript: Bacterial LPS E. coli (0.2 g/l; serotype O26:B6, Sigma-Aldrich, USA) was injected into 256 the dorsal hippocampus (DH) for induction of acute local neuroinflammation. This LPS dose was selected because pilot studies indicated that this dose of LPS induces moderate proinflammatory response in the hippocampus (data not shown).

In other studies the LPS concentrations for intrahippocampal injections were:

  • 2µg of LPS in 2µl (https://doi.org/10.1016/j.neurobiolaging.2009.01.012)
  • 2µg or 4µg of LPS in 1µl (https://doi.org/10.1016/S0197-4580(01)00292-5)
  • 1, 2.5 and 5 μg/rat of LPS, the volume of injection was 1 µl (https://doi.org/10.1007/s00221-013-3415-6)
  • 4µg/ µl (https://doi.org/10.1371/journal.pone.0039656)
  • 4 μg/μl LPS (https://doi.org/10.1002/glia.20272)
  • 2µg or 4µg of LPS in 1µl (http://dx.doi.org/10.1016/j.bbrc.2016.09.073)
  • 10µg/ µl of LPS, the volume of injection was 0.3 µl (http://dx.doi.org/10.1016/j.bbi.2014.01.017

Round 2

Reviewer 2 Report

It is the reviewer's view that protein expression studies must be done to support the gene expression results.

There should be the quantification of expression in the images to compare the results in different groups.